# Small Peptides: Orchestrators of Plant Growth and Developmental Processes

**DOI:** 10.3390/ijms25147627

**Published:** 2024-07-11

**Authors:** Shuaiqi Lu, Fei Xiao

**Affiliations:** Xinjiang Key Laboratory of Biological Resources and Genetic Engineering, College of Life Science and Technology, Xinjiang University, Urumqi 830046, China; 107552301026@stu.xju.edu.cn

**Keywords:** small peptides, peptide hormones, small secreted peptides, peptide–receptor module, plant growth and development

## Abstract

Small peptides (SPs), ranging from 5 to 100 amino acids, play integral roles in plants due to their diverse functions. Despite their low abundance and small molecular weight, SPs intricately regulate critical aspects of plant life, including cell division, growth, differentiation, flowering, fruiting, maturation, and stress responses. As vital mediators of intercellular signaling, SPs have garnered significant attention in plant biology research. This comprehensive review delves into SPs’ structure, classification, and identification, providing a detailed understanding of their significance. Additionally, we summarize recent findings on the biological functions and signaling pathways of prominent SPs that regulate plant growth and development. This review also offers a perspective on future research directions in peptide signaling pathways.

## 1. Introduction

Small peptides (SPs), consisting of 5 to 100 amino acids, are widely distributed in plants and animals and function as critical signaling molecules regulating growth, development, and stress responses [1,2]. Most SPs are encoded by precursor genes, with typical peptide precursors having conserved family-specific sequences at the C-terminus [3]. These precursors undergo proteolytic cleavage and post-translational modifications to yield mature, functional peptides [4]. Mature SPs are recognized by nearby cell membrane surface receptors, such as receptor kinases and receptor-like kinase families. Alternatively, SPs may undergo long-distance transport to bind to the extracellular domains of corresponding receptors [5,6]. This binding event activates the intracellular protein kinase domains of the receptors, triggering downstream signaling cascades such as MAPK signaling and transcription factors [7,8,9].

SPs are essential regulators of plant growth and development, influencing cell proliferation in stems, roots, leaves, flowers, fruits, and seeds [10,11,12]. They also mediate plant resistance to environmental stresses such as drought, salt, pests, and diseases [8,13,14]. Numerous SPs have been identified in various crops beyond model plants like Arabidopsis and rice, with their homologous genes and structures showing a high conservation across plant species [3,15].

This review provides an overview of the latest methods for identifying peptides, which involve screening all amino acid sequences and small open reading frames (sORFs) of a species to pinpoint candidate peptides. We then delve into SPs that are intricately linked to plant growth and development, focusing on peptide families such as phytosulfokine (PSK), plant peptide-containing sulfated tyrosine (PSY), Casparian strip integrity factors (CIF), root growth factor (RGF), clavata/embryo surrounding region (CLE), epidermal patterning factor/epidermal patterning factor-like (EPF/EPFL), LURE, rapid alkalinization factor (RALF), C-terminally encoded peptide (CEP), and inflorescence deficient in abscission (IDA)/IDA-LIKE (IDL). This review explores their origins and characteristics and summarizes how they fine-tune plant growth and development by activating various downstream signal transduction pathways. Additionally, it offers a perspective on future research directions in peptide signaling pathways, aiming to inspire further exploration and discovery in this dynamic field.

## 2. Classification and Identification of SPs

SPs, known as small molecule signaling peptides, exhibit remarkably conserved homologous gene structures and domains among plants [2]. The first small plant peptide, systemin, was identified in *Solanum lycopersicum* and regulates the resistance response to wounds caused by pests and diseases [16]. Since then, various plant SPs have been identified and annotated using classical biochemical and molecular methods. Additionally, many predictive tools and bioinformatics approaches for identifying SPs are emerging.

### 2.1. Classification Based on Origin

Most mature SPs consist of approximately 5–30 amino acids. They can be categorized into two main groups based on their origin: precursor-derived and non-precursor-derived peptides. Precursor-derived peptides are processed from precursor proteins, often with a signal peptide at the N-terminus guiding their maturation into functional peptides [2,6]. These peptides can be further classified into functional and non-functional precursors. For example, CAP-derived peptide (CAPE) and Subtilase peptide (SUBPEP) are functional precursors involved in salt tolerance, while non-functional precursors lack a defined biological role, typically arising from extended preproproteins [17,18,19]. Non-functional precursors can be categorized into three groups based on their mature peptide characteristics (Figure 1).

Peptides incorporating posttranslational modifications (PTMs), such as proline hydroxylation, hydroxyproline arabinosylation, and tyrosine sulfation, which confer biological activity and chemical stability [20]. Examples include PAMP-induced secreted peptides (PIP), CEP, IDA/IDL, and CLE peptides.Cysteine-rich peptides (CRPs) containing structural domains with 2–16 Cys residues influence peptide structure and activity via intramolecular disulfide bond formation [2,21]. The EPF/EPFL peptide family is a notable example of regulating plant stomata development [22].Non-cysteine-rich/non-PTM peptides play roles in plant defense responses. Examples include systemin (SYS), plant elicitor peptide (PEP), and plant natriuretic peptides (PNP) [23].

Nonprecursor-derived peptides are directly translated from sORFs without intermediate propeptides or additional processing steps [6]. These peptides can be classified into three classes based on the genomic location of the sORF genes (Figure 1): (1) peptides encoded by upstream ORFs on the mRNA 5’ end, (2) peptides encoded within longer non-coding proteins in other transcripts, and (3) peptides encoded by sORFs within primary transcripts of microRNAs (pri-miRNAs). In summary, SPs display diverse characteristics and functions in plants. Understanding the origin and classification of SPs provides insights into their regulatory roles in plant growth, development, and stress responses.

### 2.2. Classification Based on the N-Terminal Sequence

Based on the variation in the N-terminal sequence of the peptide precursor protein, plant SPs can be categorized as non-secretory and secretory peptides [6,24]. Non-secretory peptides can regulate intracellular processes and are released into the extracellular matrix from injured cells to activate plant defense responses directly [25]. For example, PEP is a typical non-secretory peptide that triggers plant defense responses against pathogens [25]. Recent research has also revealed that PEP/REF1 functions as a systemin-independent local wound signal in tomato plants, predominantly orchestrating localized defense mechanisms and regenerative processes in response to injury [9]. Additionally, it enhances the regenerative capacity of callus tissue [9,26].

In contrast, secretory peptides are synthesized intracellularly and transported to the extracellular space [6,27]. They often exert their effects through active transport, traversing the epidermis or xylem, facilitating intercellular signal transduction, and regulating the activities of adjacent cells in the extracellular milieu [3,27]. Peptides incorporating post-translational modifications (PTMs) and cysteine-rich peptides (CRPs) are classified as secreted peptides. Importantly, PTMs represent the most prevalent type of plant peptides [24]. The precursor protein transforms into a mature peptide through electrostatic charge, hydrophilicity, and conformation modifications. The mature peptide then binds to receptors, activating downstream signal transduction pathways [21,24].

### 2.3. Identification Methods for SPs

Various functional plant SPs that regulate plant growth have been identified and annotated [6,28]. Traditional biochemical and genetic approaches have been used to validate the biological functions of plant peptides. However, these methods are not ideal for high-efficiency peptide identification, often yielding only a few identified peptides [29].

Currently, researchers have integrated novel peptidomics, bioinformatics screening, and biochemical and genetic validation methods for peptide identification, overcoming the high costs and technical difficulties associated with traditional biochemical analysis and circumventing obstacles arising from low gene abundance and redundancy [2,30]. For instance, high-throughput analytical approaches that couple selective enrichment, fractionation, phenotype screening, and mass spectrometry identification provide an established framework for screening plant tissues for biologically relevant small secreted peptides (SSPs) [30,31]. This method begins with selective enrichment strategies to isolate molecules of interest from complex crude extracts, commonly using techniques such as size exclusion ultrafiltration [32], gel-based separations, solvent extractions, and size exclusion chromatography (Figure 2A).

Next, further fractionation is performed based on physicochemical properties (e.g., polarity, hydrophobicity, stability, solubility). SSPs can be screened for bioactivity in cell-based or cell-free systems to evaluate functions such as antimicrobial activity or free radical scavenging [33]. After detecting fractions with relevant bioactivity, high-throughput LC-MS/MS is used to sequence unknown SSPs [31]. However, mass spectrometry identification of SSPs faces challenges due to insufficient SSP representation in protein databases and unclear mechanisms of SSP maturation.

A recent bioinformatics approach for peptide identification involves screening for SSPs within the entire proteome of a given plant species. Initially, the focus is on the amino acid sequences of the proteins. Proteins containing fewer than 250 amino acids are selected as potential secretory peptides, as they typically possess N-terminal signal peptides. Subsequently, signal peptide prediction is performed on these small peptides, eliminating those lacking N-terminal signal peptides. Since secretory peptides lack transmembrane domains, further prediction of transmembrane domains is employed on the remaining peptides, removing those exhibiting such structures. Finally, peptides with C-terminal K/HDLE sequences, which serve as endoplasmic reticulum docking proteins, are discarded, yielding the secretory peptide sequences for the given species (Figure 2B) [21,24].

Notably, genes encoding SSPs are often overlooked during genome annotation due to their short lengths [2]. Therefore, a more comprehensive approach involves screening sORFs to identify SSPs (Figure 2B). This method first identifies non-coding sequences (NCDS) from the chromosomes using software tools like sORF finder. The subsequent steps are similar to the methods described earlier, wherein the presence or absence of an N-terminal signal peptide, the presence of transmembrane domains, and whether the sORF encodes an ER-docking protein are employed as distinguishing SSP characteristics for selecting potential sORFs within the NCDS (Figure 2B). It is worth noting that only a tiny fraction of sORFs can be categorized into known gene families, while most remain unclassifiable. This issue may be attributed to undiscovered peptide families and the lack of comprehensive sORF annotation, which could lead to false positives. Together, this represents a novel approach to plant peptide identification, highlighting the potential of bioinformatics in unraveling SSPs. This method offers a promising direction for future research, utilizing bioinformatics to identify and characterize secretory peptides, thereby advancing our understanding of their roles in plant biology.

## 3. Crucial SPs in Regulating Plant Growth and Development

Many SPs that play regulatory roles in plant growth and development have been identified. Among these, critical SPs include sulfated peptides, CLE peptides, and several other types. These peptides mediate growth and developmental processes in various parts of plants by interacting with different receptors (see Table 1 for details).

### 3.1. Tyrosine-Sulfated Peptides

Sulfated peptides are plant hormone-like molecules known for their potent activity, even at nanomolar concentrations. Four distinct sulfated peptides have been characterized in plants—PSK, PSY, CIF, and RGF. These peptides mediate growth and development processes by activating specific signal transduction pathways.

#### 3.1.1. PSK Peptides

PSK peptides, originating from 80 to 120 amino acid preproproteins, were first identified in asparagus suspension cell cultures as growth promoters [33,63,64]. They are widely distributed across plants, including angiosperms and gymnosperms, with several members in various species sharing significant sequence similarity and a conserved C-terminal domain [11,34,65,66]. PSK maturation involves tyrosine sulfation and proteolytic cleavage, facilitated by plant-specific enzymes such as tyrosylprotein sulfotransferases (TPSTs) and subtilisin serine proteases [67,68].

PSK peptides play crucial roles in various aspects of plant growth [13,63,66]. Synthetic PSK-α peptide treatment significantly enhances root and hypocotyl elongation in Arabidopsis, mediated by the PSK receptor (PSKR) [69,70]. Loss-of-function mutants of *PSKR* and *AtTPST* exhibit defects in root growth and cell size reductions, confirming the importance of PSKs in promoting growth [35,69,71]. BRI1-associated receptor kinase 1 (BAK1)/somatic embryogenesis receprot-like kinases (SERKs) act as co-receptors in the PSK signaling pathway; they are not directly involved in the binding of PSK [70,72]. Furthermore, PSK regulates plant reproductive development, promoting quorum sensing during pollen germination and pollen tube growth [34,73,74,75]. Specific *PSK* expression patterns in soybean seeds and overexpression of *PSK* genes in wheat enhance seed and grain growth, respectively [4,66].

PSK also modulates fruit ripening and nutrient accumulation, with exogenous PSK application promoting tomato fruit ripening via PSKR-dependent pathways (Figure 3A) [35]. Exogenous application of SlPSK5 enhances tomato fruit ripening, but this effect is lost in *pskr1* mutants, indicating PSKR dependency [35]. PSK also impedes the ubiquitination alteration of its receptor by PUB12/13, amplifying the stability of PSKR1 in tomatoes (Figure 3A) [76]. Activation of the SlPSK5–SlPSKR module phosphorylates the transcription factor DREBF2, boosting the expression of ripening-related genes (Figure 3A) [35]. A recent investigation elucidated that PSK orchestrates the equilibrium between growth and defense mechanisms in tomatoes by phosphorylating distinct amino acid residues of GLUTAMINE SYNTHETASE 2 (GS2) via the action of calcium-dependent protein kinase 28 (CPK28) (Figure 3A) [64]. In summary, PSK peptides are critical in root development, pollen germination, and fruit ripening processes.

#### 3.1.2. PSY Peptides

PSY peptides were initially discovered in Arabidopsis suspension culture cells. In Arabidopsis, nine *PSY* genes encoding PSY precursor proteins with lengths ranging from 71 to 104 amino acids have been identified [12,36]. The mature PSY1 peptide comprises 18 amino acids, featuring sulfation modification on Tyr2 and hydroxylation modifications on Pro16 and Pro17 [77]. Recent studies have revealed that PSY–PSYR signaling is a switch to mediate the trade-off between plant growth and stress response [12]. PSYR1, PSYR2, and PSYR3 act as redundant negative regulators of plant growth in the absence of the PSY peptide, while PSY peptides suppress PSYR signaling, promoting root growth (Figure 3B) [12]. PSYR1 can intricately interact with and enhance the activity of AHA2 by phosphorylating its T881 residue. Activation of the plasma membrane H^+^-ATPase AHA2 by PSY1 occurs in a PSYR1-dependent manner, leading to extracellular acidification and promoting hypocotyl elongation (Figure 3B) [78,79]. Thus, the growth-promoting function of PSY family peptides may represent a delicate balance between growth and stress resilience [12,36]. Overall, the interaction between PSY and PSYR is crucial in plant growth and development.

#### 3.1.3. CIFs and TWS1 Peptides

CIFs play a crucial role in regulating the integrity of the plant Casparian strip [37,80]. In Arabidopsis, five *CIF* genes—*CIF1*, *CIF2*, *CIF3*, *CIF4*, and *TWS1*—encode post-translationally modified peptides. Their receptors, GASSHO 1 (GSO1) and GSO2, belong to the leucine-rich repeat receptor-like kinase family [81]. CIF1 and CIF2 share 21 conserved amino acids at the C-terminus and contain sulfation and hydroxylation sites that are crucial for binding to GSO1 [80,82]. The structure of the GSO1–CIF2 complex reveals an interaction that stabilizes the receptor/peptide binding, forming the GSO pathway (Figure 3C) [82,83]. After sulfation by tyrosylprotein sulfotransferases (TPSTs) in the vascular bundle, CIF1 and CIF2 are transported to endodermal cells, where they bind to the receptor complex GSO1 and SERK family proteins (Figure 3C) [84]. CIF transport halts upon Casparian strip completion, terminating the GSO pathway and lignin synthesis. Additionally, the GSO pathway regulates CASP structural domain formation and suberin synthesis in cork tissue [82,85,86].

CIF3 and CIF4 are crucial in regulating pollen development; CIF3 and CIF4 precursor proteins require cleavage by the protease SBT5.4 [38]. SBT5.4 is expressed in pollen. Mature CIF3 or CIF4 molecules are detectable by GSO1/GSO2 receptors, thereby initiating the process of pollen wall development (Figure 3C) [38]. TWS1, expressed in the embryo, exhibits weaker sequence similarity to the core region of CIF1 [20,39]. Its precursor is directly processed at the C-terminus by the ALE1 peptidase, allowing it to cross the cuticle barrier and re-enter the embryo [20]. It interacts with GSO1 and GSO2 receptors to establish cell–cell connections during embryonic epidermis formation (Figure 3C) [87]. However, few studies have examined the processing and modification of CIF peptides in plants, indicating a need for further comprehensive research.

#### 3.1.4. RGF/GLV/CLEL Peptides

The RGF peptides, also known as GOLVEN (GLV) or the CLE-like (CLEL) peptide family, consist of 13 amino acids and originate from precursor peptides of approximately 100 amino acids [88,89]. These sulfated peptides typically feature the characteristic DY-motif, except for GLV9, and a highly conserved hydroxylated proline residue. Across plant species, RGFs exhibit a combination of conserved and non-conserved amino acids [89]. Biologically active RGF peptides undergo tyrosine sulfation and proteolytic cleavage [89].

RGFs play a crucial role in root development by regulating the apical meristem. Recent studies have shown that in vitro treatments with GLVs and RGFs can significantly inhibit lateral root formation by impeding auxin accumulation during lateral root initiation [90]. Specifically, RGF peptides modulate root apical meristem activity by regulating the expression of PLETHORA (PLT) proteins, which serve as master regulators of root formation (Figure 3D) [91,92]. Most *RGF* genes are expressed in quiescent center cells and adjacent cells in roots, leading to the anticipation of peptide diffusion in the meristematic region with a gradient [93,94]. This gradient of RGF peptides delineates the gradient of PLT protein, modulating protein stability to ensure resilient root growth and development in dynamic environments [89]. Researchers have found that the RGFs and their receptor pairs (RGF–RGFR) orchestrate the concentration gradient of the PLT protein to maintain root apical meristem activity (Figure 3D) [40,91].

In Arabidopsis, three RGF receptors (RGFRs), namely RGFR1, RGFR2, and RGFR3, have been identified in proximal meristematic tissues [95,96]. It has been documented that the receptor kinases RGI4/SKM2 and RGI5 are involved in RGF perception, in conjunction with RGI1/RGFR1, RGI2/RGFR2, and RGI3/RGFR3 [94,95,96]. The quintuple mutant *rgi1,2,3,4,5* shows a significant decrease in root apical meristem activity and complete insensitivity to RGF [94]. Moreover, it has been confirmed that *RGI4/SKM2*, a paralog of RGI3/RGFR3, directly binds to RGF (Figure 3D) [10]. Subsequent studies have revealed that BAK1 and its paralogs, the SERKs, act as coreceptors of RGI1 for sensing RGF1 (Figure 3D) [10,97]. Upon recognition by the RGI1–SERKS complex at the cellular periphery, the RGF1 signal is relayed to PLTs through a YODA–MKK4/5–MPK3/6 signaling cascade (Figure 3D) [41,98]. However, the direct interaction between RGI5 and RGF has not yet been demonstrated. These findings underscore the pivotal role of the RGF–RGFR module in root development by regulating the apical meristem.

### 3.2. CLE Peptides

CLE peptides, crucial plant signaling molecules, interact with membrane-bound receptors, modulating transcription factors and phytohormone pathways [42,99]. Notably, specific CLE peptides like CLV3, TDIF, CLE40, and CLE45 play vital roles in plant development and response to stimuli [5,100]. Subsequent sections of this review will delve deeper into their regulatory roles and signaling mechanisms in plant growth and development.

#### 3.2.1. CLV3

The CLAVATA3 (CLV3) peptide, crucial for cell fate determination in the shoot apical meristem of Arabidopsis, has two mature forms identified via mass spectrometry: a 12-amino-acid peptide with hydroxyproline (Hyp) residues at positions 4 and 7, and a 13-amino-acid arabinosylated form with a histidine at position 13 [101,102]. These mature CLE peptides are recognized by the CLV1/CLV2 receptors, which consist of a leucine-rich repeat receptor-like kinase (LRR-RLK) and a leucine-rich repeat receptor-like protein (LRR-RLP) lacking a kinase domain [103,104]. Additionally, the receptors CORYNE (CRN) and RECEPTOR-LIKE PROTEIN KINASE 2 (RPK2) are involved in CLV3 signaling [43,105]. Recent studies show that CLV1, CLV2, and CRN form a receptor complex for CLV3 (Figure 4) [106].

RPK2, also known as TOADSTOOL 2 (TOAD2), mediates the CLV3 signaling cascade by associating with BAM1 [105,107,108]. The three BARELY ANY MERISTEM (BAM) receptor kinases, clustered within the same clade as CLV1, play a pivotal role in maintaining stem cell preservation along the peripheries of the shoot apical meristem [106,109]. BAM1 and BAM2 can bind to CLV3, functioning synergistically with CLV1 as supplementary receptors for CLV3. This interaction initiates a downstream phosphorylation cascade via mitogen-activated protein kinase (MAPK) pathways, with MPK3 and MPK6 phosphorylation ultimately regulating stem cell equilibrium (Figure 4) [110]. The CLV3 signal is perceived by multiple receptor complexes—CLV1/CLV1 homomeric, CLV2/CRN heteromeric, and possibly RPK2 homomeric—negatively regulating WUSCHEL, a master regulator in plant growth signaling, thereby controlling the shoot apical meristem (Figure 4) [106,108].

#### 3.2.2. TDIF

TDIF, initially derived from *Zinnia elegans*, regulates plant vascular meristem development [111]. In Arabidopsis, the *CLE41* and *CLE44* genes share a 12-amino-acid sequence within their CLE domains, with CLE44 resembling TDIF but lacking glycosylation [29]. Treatment with TDIF inhibits xylem cell differentiation while promoting procambial cell proliferation, highlighting its role in plant development [44]. The receptor for TDIF, TDR, belongs to the LRR-RLK XI subfamily and plays a pivotal role in maintaining the vascular meristem [112,113,114]. In a manner parallel to the CLV3 signaling pathways, the WUS family protein WUS-RELATED HOMEOBOX 4 (WOX4) acts as a mediator of TDIF signaling by facilitating the proliferation of procambial cells (Figure 4) [45]. Downstream effectors of the TDIF pathway, glycogen synthase kinase 3 proteins (GSK3s), suppress xylem differentiation via the WOX4 pathway. TDR activates GSK3s at the plasma membrane in a TDIF-dependent manner [45]. Inhibition of GSK3s induces xylem cell differentiation through the transcription factor BRI1-EMS SUPPRESSOR 1 (BES1) [45,115]. Additionally, a NAC domain transcription factor, XVP, localizes on the plasma membrane, interacting with the TDIF co-receptor PXY-BAK1 to form a complex (Figure 4) [113,114]. *XVP* expression is localized in the cambium and is critical in regulating xylem differentiation and vascular patterning [114]. Subsequent investigations have elucidated that XVP governs xylem differentiation via the pivotal factor VASCULAR-RELATED NAC-DOMAIN6 (VDN6). Notably, the conservation of the CLE41/TDIF–TDR–WOX4 signaling pathway in wood-forming Populus species suggests its role as a universal regulator of vascular development (Figure 4) [116].

#### 3.2.3. CLE40

CLAVATA3/ESR-RELATED40 (CLE40), the closest homolog of CLV3 in Arabidopsis, regulates columella stem cells (CSCs) in the root meristem [46]. CSCs, located at the root apices, perceive gravitational stimuli and have a unique cell wall composition that protects the root meristem [46]. *CLE40* expression in CSCs controls cell division, as evidenced by *cle40* mutants displaying excessive CSC layers [117]. Genetic studies indicate that CLE40 functions through the receptor kinases ARABIDOPSIS CRINKLY4 (ACR4) and CLAVATA1 (CLV1) to regulate *WOX5* expression (Figure 4) [118,119]. ACR4, but not CLV1, is transcriptionally upregulated following ectopic CLE40 peptide treatment [120,121]. ACR4 is also implicated in forming the epidermal cell layer and initiating lateral roots (Figure 4) [118]. CLE40’s action is limited to cortex cells, whereas *WOX5* expression in the quiescent center positively regulates stem cell production, suggesting a balance between CLE40 and WOX5 in regulating the stem cell population of the root meristem (Figure 4) [119,122].

#### 3.2.4. CLE45

The interaction between CLE45 and BAM3 establishes a signaling module that regulates vascular development independently of the TDIF–TDR pathway [47,123]. Synthetic CLE45 peptide inhibits root growth and protophloem differentiation in Arabidopsis [48]. The class II LRR-RLK, known as CLE-RESISTANT RECEPTOR KINASE (CLERK), also termed CIK2, is indispensable for the complete perception of CLE45 and other CLE peptides active in early proto-phloem development [124]. However, CLERK operates within a genetically distinct pathway from BAM3 and CLV2/CRN. Notably, membrane-associated kinase regulator 5 (MAKR5) emerges as a pivotal facilitator of CLE45 signaling, operating downstream of BAM3. CLE45 signaling recruits MAKR5 to the plasma membrane and promotes its accumulation in developing sieve elements (Figure 4) [48]. This signaling module interacts genetically with the plasma membrane-localized proteins BREVIS RADIX (BRX) and OCTOPUS (OPS), which are critical for sieve–element (SE) cell fate control [125,126]. Deletion of BRX or OPS leads to defecting in protophloem differentiation due to interference with interactions between CLE45 signaling receptors BAM3 and the CLV2/CRN complex. *brx* and *ops* mutants also exhibit abbreviated root characteristics and increased lateral root production from primary root branching zones [125,126]. The CLE45–SKM1/SKM2 pathway facilitates intercellular communication between male and female cells, aiding pollen tube growth under heat stress and fertilization [48]. While the regulatory role of CLE in root and pollen tube development is documented, the detailed molecular framework and specific signaling components governing these processes remain unclear.

In addition to the previously mentioned CLE peptides, various others exhibit diverse functionalities. For instance, the peptides CLE1, 3, 4, and 7 suppress lateral root formation via the CLV1 receptor under nitrate deficiency [127]. CLE18 and CLE26 are associated with root growth [88,128]. The CLE5 and CLE6 complexes influence B-type CLE and shoot development [129]. CLE8 regulates embryo and endosperm development in *Brassica napus* [130], while CLE10 influences protoxylem vessel formation [131]. Overall, CLE peptides are crucial in various aspects of plant growth and development.

### 3.3. Other SPs

#### 3.3.1. EPF/EPFL Peptides

The EPF/EPFL peptides are characterized by abundant cysteine residues and an N-terminal signal peptide [22]. Upon proteolytic cleavage, mature EPF/EPFL peptides feature a conserved C-terminal region, typically containing six or eight cysteine residues that facilitate intramolecular disulfide bond formation [132,133]. EPF1 and EPF2, which share significant homology, interact with the ERECTA-LIKE1 (ERL1) and ERECTA-LIKE2 (ERL2) receptors, respectively [49,50]. EPF/EPFL peptides are known to often negatively regulate stomatal density, thereby inhibiting stomatal development [22,132]. Additionally, several Arabidopsis EPF members, including EPF1, EPF2, and EPFL9, regulate stomatal development by binding to a complex formed by TMM (too many mouths) and ERECTA (ER) receptor kinases (Figure 5A) [134]. EPFL9 (STOMAGEN) competes with EPF2 for ER receptor binding and fine-tuning stomatal development (Figure 5A) [51,134]. In contrast, *CHALLAH (CHAL)/EPFL6* exhibits low expression in leaves and the stomatal lineage [52]. Although single *chal* mutants and triple mutants do not display any discernible phenotype in stomatal development, CLL2/EPFL4 and CHAL/EPFL6 are predominantly expressed in the endodermis of the inflorescence stem, where they regulate proper stem architecture (Figure 5A) [22,52]. Consequently, CLL2/EPFL4 and CHAL/EPFL6 primarily act on inflorescence stem growth regulation [22]. Recent research has elucidated that EPFL4 and EPFL6 interacts with the ER–SERK receptor complex, activating the downstream MKK5/6–MPK6 cascade, thereby dynamically regulating cellular proliferation in stamen filaments (Figure 5A) [135]. Despite the incomplete understanding of EPF/EPFL-associated signaling pathways, research suggests that many EPF peptides exert a negative regulatory effect on stomatal density, thereby inhibiting stomatal development [22,135].

#### 3.3.2. LURE Peptides

LUREs, the first identified pollen-tube attractants found in *Torenia fournieri*’s synergid cells, belong to the defensin-like cysteine-rich peptide family [136]. Among the sixteen cysteine-rich peptides isolated from these cells, TfCRP1 and TfCRP3, known as LURE1 and LURE2, act as pollen tube attractants [53]. Despite variations in their sequences, all LUREs share a conserved pattern of six cysteine residues for disulfide bond formation. However, comparisons across species, such as Torenia and Arabidopsis, reveal that the alignment of cysteine residues is the only conserved feature [53,136].

A recent study identified pollen-specific receptor kinase 6 (PRK6) as a crucial receptor for sensing the attractant peptide AtLURE1 (Figure 5B) [137]. It was observed that pollen tubes from *prk* loss-of-function mutants could not respond to AtLURE1 [137]. PRK6, consisting of six leucine-rich repeats (LRRs), belongs to a subclade containing eight PRK family receptors (PRK1–8) in Arabidopsis [137,138]. Interestingly, studies on PRK family receptors in tomatoes and Arabidopsis have indicated their functional roles in regulating the efficiency of pollen tube growth [139]. The results suggest that *prk* mutants (*prk3, prk6, prk8*, *prk1*, *prk3*, *and prk6*) demonstrate a dramatic decrease in pollen tube growth and fertility, while individual *prk* mutants do not exhibit such pronounced effects [139]. Although the exact relationship with PRKs remains unclear, a separate group of pollen-expressed RLKs, including MALE DISCOVERER1 (MDIS1), MDIS2, MDIS1-interacting receptor-like kinase 1 (MIK1), and MIK2, have also been implicated in the sensing of AtLURE1 (Figure 5B) [138,139]. Even though the downstream signaling pathway of LUREs remains incompletely elucidated, we can still infer that PRK6, in conjunction with the multiple receptor components mentioned earlier, may serve as a pivotal player in regulating pollen tube growth and attraction by detecting extracellular ligands such as AtLURE1.

#### 3.3.3. RALF Peptides

RALF was initially discovered in tobacco plants for its ability to raise the pH of the culture medium quickly [140]. RALFs are widespread in various plant species, including dicots, monocots, and gymnosperms. In Arabidopsis, 40 *RALF* and *RALF-like* (*RALFL*) genes have been identified, showing diverse expression patterns in different organs and tissues [140,141]. RALF belongs to the cysteine-rich peptide family in plants and is characterized by four conserved cysteine residues forming two disulfide bonds, which are crucial for proper folding [142]. Upstream of the mature RALF N-terminus is a pair of arginine residues, serving as cleavage sites for peptidases and coordinating immune responses [142]. Downstream, the YISY motif is crucial for receptor binding. The C-terminal sequences GASYY and RCRR(S) of the RALFs might stabilize the peptide structure or mediate interactions between peptides [143].

Catharanthus roseus receptor-like kinase 1-like (CrRLK1L) proteins recognize RALFs, influencing plant growth [144]. For instance, the binding of RALF1 to FERONIA (FER), a member of the CrRLK1L family, suppresses lateral root growth, where BAK1 forms a receptor complex with FER (Figure 5C) [54,145]. Additionally, RALFs regulate plant reproductive development. RALF4 and RALF19, expressed in pollen, interact with the ANXUR1/2 (ANX1/2)-BUPS1/2 (Buddha’s paper seal 1/2) complex, influencing pollen tube growth (Figure 5C) [55]. As the pollen tube approaches the embryo sac, RALF34 competes with RALF4 and RALF19 for binding to the ANX1/2-BUPS1/2 complex, leading to pollen tube rupture and sperm release, thereby completing double fertilization (Figure 5C) [56,146]. Moreover, LORELEI-like-GPI-anchored proteins 2 and 3 (LLG2/3) serve as co-receptors alongside ANX1/2 and BUPS1/2, facilitating their function (Figure 5C) [56]. When the pollen tube grows towards the embryo sac, it was documented that RALF4/19 instigated the ANX1/2-BUPS1/2-LLG2/3-MARIS cascade, subsequently activating AtMLO1, 5, 9, and 15 (Figure 5C) [147]. This activation facilitated tip-focused Ca^2+^ influx, upholding pollen tube integrity (Figure 5C) [147]. Together, RALFs play crucial regulatory roles mediated through receptor interactions, impacting processes of root development and plant reproduction.

#### 3.3.4. CEP Peptides

CEP peptides originate from proteins featuring an N-terminal secretion signal, a variable domain, and one or more CEP domains. These peptides, each spanning 15 amino acids, are reportedly essential in plant growth and development [148,149,150]. For instance, AtCEP3 negatively regulates lateral root numbers in Arabidopsis, as demonstrated by increased lateral root density in *cep3* mutant plants compared to wild-types under various nutrient-deficient and abiotic stress conditions [57]. Additionally, CEPs are crucial in negatively regulating lateral root emergence in *Medicago truncatula*. Overexpression of *MtCEP1* reduces lateral root number, while simultaneous knockdown of *MtCEP1* and *MtCEP2* increases lateral root number (Figure 5D) [57]. CEP5 regulates the auxin response in lateral root formation by enhancing AUX/indole-3-acetic acid (IAA) stability, thereby suppressing the expression of auxin-responsive genes (Figure 5D) [58].

In recent studies, overexpression of *GhCEP46-D05* in cotton and Arabidopsis resulted in reduced plant height, fiber length, root length, and the length and width of mature seeds in transgenic lines (Figure 5D0 [59]. Notably, CEP peptides impact plant development by specifically binding to CEP RECEPTORS 1/2 (CEPR1/CEPR2) kinases [151,152]. The CEP–CEPR1 pathways inhibit lateral root growth in the presence of sucrose, other metabolizable sugars, and increased light intensity by reducing lateral root meristem size and the length of mature lateral root cells (Figure 5D) [152]. Recent studies have revealed that CEP and cytokinin signaling mediate the action of CEP DOWNSTREAM (CEPD), thereby inhibiting primary root growth (Figure 5D) [150]. Despite some literature outlining the functional roles of the CEP–CEPR signaling pathway in plant lateral root development, further elucidation is needed to understand the comprehensive regulatory network of this pathway in plant development [150,152].

#### 3.3.5. IDA/IDL Peptides

IDA/IDL peptides, a class of small signaling proteins, are generated via the proteolytic cleavage of their precursor protein catalyzed by subtilases [153]. They possess an N-terminal secretory signal peptide, guiding the protein to the extracellular milieu, while the C-terminal proline-rich extended motif, known as the EPIP motif, comprised of 20 amino acids, plays a pivotal role in inducing the abscission of floral organs post-pollination [154,155]. Reportedly, IDA/IDL peptides regulate the shedding of cauline leaves in response to dehydration stress [156]. Under water deficit conditions, when leaves show signs of wilting, the bioactive IDA peptide ligand is perceived by a receptor complex composed of either the receptor kinases HAESA or HAESA-like 2 (HSL2), along with SERK coreceptors (Figure 5E) [154,156]. This recognition event initiates a downstream signaling cascade involving MAPKs, ultimately governing the expression of cell-wall-modifying and hydrolytic enzymes, particularly polygalacturonases and xyloglucan endotransglucosylase/hydrolases (Figure 5E) [154,156]. These enzymes degrade the middle lamella, thereby inducing cell separation.

Furthermore, corroborating evidence confirms that *IDA* expression occurs shortly after abscission zone cells attain receptivity to abscission signals, signifying early loosening of cell walls [60]. Moreover, the ectopic expression of *LcIDL1* in Arabidopsis results in the enlargement of abscission zone cells at the sites where organs detach from the plant, leading to the abscission of floral organs [62]. These discoveries suggest that the IDA/IDL peptide likely orchestrates the initial relaxation of cell walls and subsequent detachment of abscission zone cells, possibly by regulating cell wall remodeling enzymes [62]. The initial cell wall relaxation primarily occurs in the root cap and inflorescence, ultimately leading to their detachment [61,157]. This finding emphasizes the vital role of IDA translocation to the apoplastic space in its functionality. In summary, IDA/IDL-mediated signaling pathways play essential roles in promoting plant organ abscission.

## 4. Conclusions and Perspectives

Small peptides, consisting of 5–100 amino acids, are crucial signaling molecules in plants, exhibiting diverse processing, folding, and modifications [84]. Advances in sequencing, informatics, molecular biology, genetics, and biochemistry have made it possible to identify and understand these peptides’ functions [32]. These peptides regulate key plant processes such as cell proliferation, tissue differentiation, organ formation, reproductive development, maturation, and senescence, and they are vital for responding to environmental stresses. Despite progress, challenges remain due to their short length and low abundance. Recent advancements in genomics and transcriptomics have expanded research from model plants like Arabidopsis and rice to economically important crops, leading to the identification and characterization of more functional peptides [158,159].

In conclusion, small plant peptides are a promising emerging research area. The research methodologies focused on SPs require further development, particularly in elucidating the mechanisms the receptor–ligand interactions of new SPs and their functional roles. Future developments could enable the quantitative production of plant peptide hormones to regulate growth, enhance stress resistance, improve yields, and address challenges in eco-agriculture. Continued innovation in this field could revolutionize agriculture and significantly contribute to sustainable food production.

## Figures and Tables

**Figure 1 ijms-25-07627-f001:**
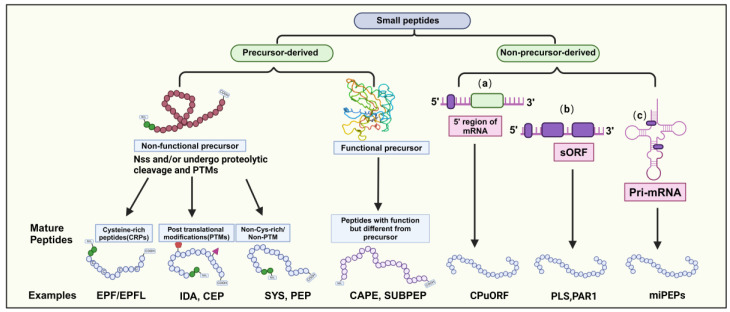
Classification of Small Peptides in Plants. Small peptides (SPs) in plants fall into two categories: precursor-derived and non-precursor-derived. Precursor-derived peptides can be functional or nonfunctional; functional ones exhibit their role post-transport, while nonfunctional ones may undergo post-translational modifications and are categorized as Cys-rich or non-Cys-rich/non-PTM. Non-precursor-derived peptides are encoded by small open reading frames (sORFs) containing fewer than 100 amino acids, typically found in (a) the 5′ upstream region of the main open reading frame (ORF), (b) transcripts not exceeding 100 amino acids, or (c) primary transcripts of microRNAs (pri-miRNAs).

**Figure 2 ijms-25-07627-f002:**
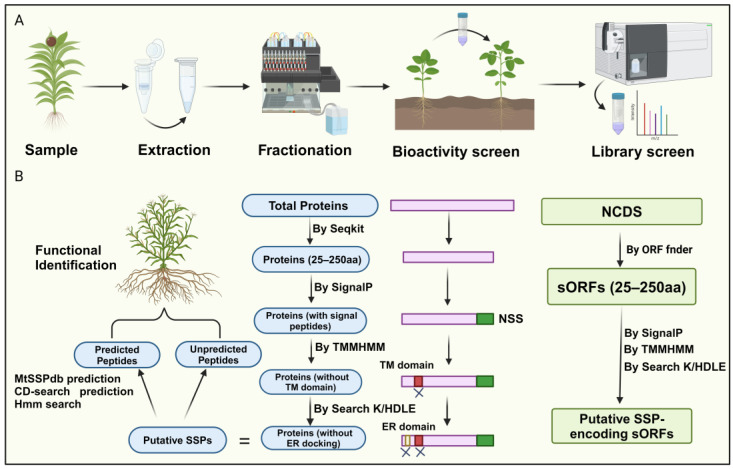
Identification of small peptides in plants. (**A**) The workflow includes four steps: extracting and enriching the low-molecular-weight (MW) fraction of the secreted proteome, fractionating or isolating low-MW fractions, testing SSP bioactivity, and analyzing SSP fraction libraries using high-resolution/high-mass accuracy LC-MS/MS with de novo search strategies. (**B**) Recent bioinformatics methods for peptide identification follow a comprehensive approach. Initially, proteins are predicted based on size criteria, with SignalP predicting N-terminal signal sequences (NSS) and TMMHMM identifying and excluding transmembrane domains. ER domains are targeted to eliminate proteins associated with endoplasmic reticulum docking. This process identifies putative SPs for further validation. Similarly, a similar strategy is applied to identify noncoding small secreted peptides (NCDS), initially identifying potential SSP-encoding small open reading frames (sORFs) and validating them using similar predictive tools.

**Figure 3 ijms-25-07627-f003:**
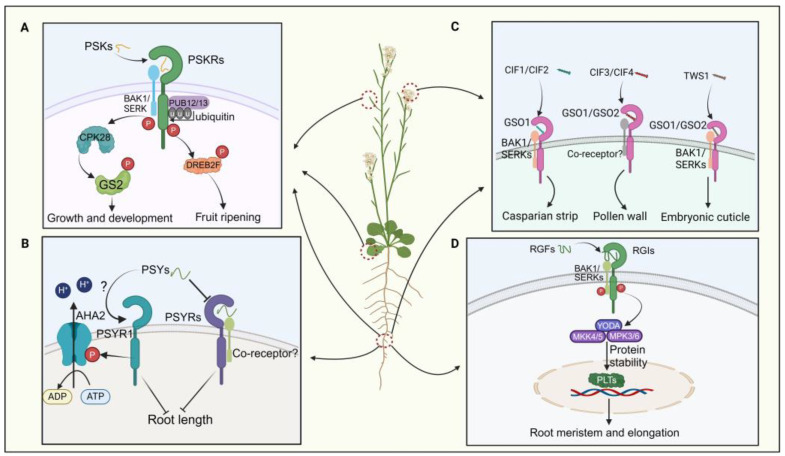
The signaling pathways of four sulfated peptides. (**A**) The PSK signaling pathway regulates plant growth, development, and fruit ripening. (**B**) The interaction between PSYs and PSYR governs the regulation of root elongation. (**C**) CIFs interact with the CSOs-BAK1 co-receptor to modulate the formation of the plant’s Casparian strip, pollen wall development, and embryonic cuticle formation. (**D**) The RGF signaling pathway modulates root meristem development and elongation.

**Figure 4 ijms-25-07627-f004:**
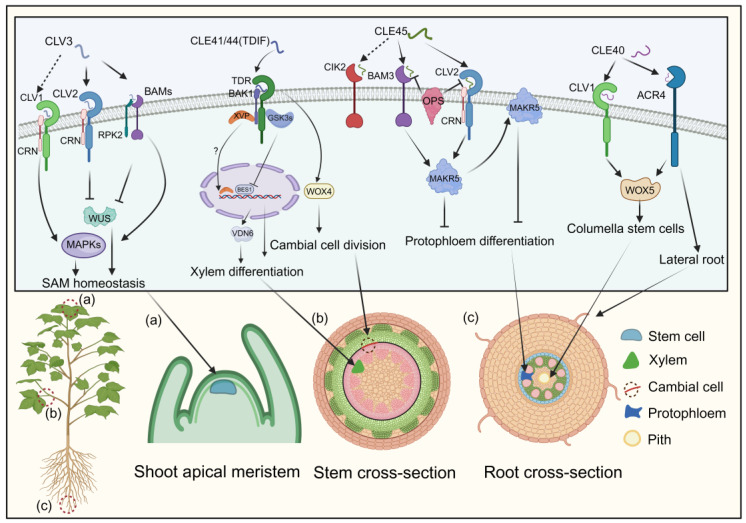
The processes modulated by CLE signaling pathways and their impact on cells. CLEs bind to receptors, including CLV2, CLV3, BAMs, TDR, CIK2, and ACR4, triggering the recruitment of co-receptors CRN, RPK2, and BAK1 to the complex. These initial interactions initiate downstream signaling cascades regulating SAM homeostasis, xylem differentiation, cambial cell division, protophloem differentiation, columella stem cells, and lateral root development. Arrows and bars represent positive and negative regulation, while solid and dashed lines denote direct and indirect regulation.

**Figure 5 ijms-25-07627-f005:**
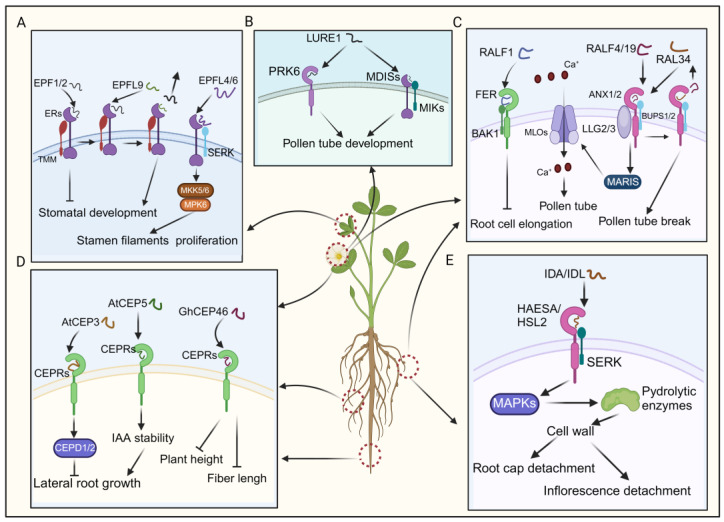
The signaling transduction pathways and functional implications of EPF/EPFL, CEP, RALF, LURE, and IDA/IDL. (**A**) EPFs modulate stomatal development by binding to the ER–TMM co-receptor and regulate stamen filament proliferation by interacting downstream with the ER–SERK co-receptor. (**B**) The binding of LURE1 to its receptor promotes the growth of pollen tubes. (**C**) Activation of downstream signaling pathways via the binding of RALFs to their receptors regulates the development of pollen tubes and the elongation of root cells. (**D**) AtCEP3 and AtCEP5 interact with CEPRs to initiate signaling cascades, thereby modulating lateral root growth. GhCEP46 binds to CEPRs to inhibit fiber length and plant height. (**E**) IDA/IDL’s binding with the HSL2–SERK co-receptor triggers downstream signal transduction, facilitating root cap and inflorescence detachment.

**Table 1 ijms-25-07627-t001:** Small peptides involved in plant growth and development.

Peptide	Plant	Receptor	Function	References
PSKs	*Arabidopsis*, wheat	PSKR1, PSKR2	Regulate root growth, cell size, pollen germination, pollen tube growth, seed development.	[4,34]
PSK5	Soybean	PSKR1	Promotes fruit ripening and nutrient accumulation	[35]
PSYs	*Arabidopsis*	PSYR1, PSYR2, PSYR3	PSY promotes root growth by binding to the PSYR receptors	[12,36]
CIF1/2	*Arabidopsis*	GSO1	Regulate casparian strip development	[37]
CIF3/4	*Arabidopsis*	GSO1/GSO2	Regulate pollen development	[38]
TWS1	*Arabidopsis*	GSO1/GSO2	Participate in embryonic cuticle development	[39]
RGF1	*Arabidopsis*	RGI1, RGI2, RGI3, RGI4	Regulates root development by modulating the apical meristem	[40,41]
CLV3	*Arabidopsis*	CLV1, CLV2, BAMs	Maintains the proper differentiation of stem apical meristem cells	[42,43]
CLE41/44, TDIF	*Arabidopsis*	TDR	Enhances cambial cell division and xylem differentiation	[29,44,45]
CLE40	*Arabidopsis*	CLV1, ACR4	Regulates columella stem cells in the root meristem	[46]
CLE45	*Arabidopsis*	CIK2, BAM3, CLV2	Inhibits root growth and protophloem differentiation	[47,48]
EPF1/2	*Arabidopsis*	ERL1, ERL2	Inhibit stomatal development	[49,50]
EPFL9	*Arabidopsis*	ERL1, ERL2	Regulates stomatal development	[51]
EPFL4/6	*Torenia fournieri*, *Arabidopsis*	ER	Regulate inflorescence and stem growth	[52]
LURE1/2	*Arabidopsis*	PRK6, MDIS	Regulate pollen tube growth	[53]
RALF1	*Arabidopsis*	FER	Inhibits lateral root growth	[54]
RALF4/19, RALF34	*Arabidopsis*	ANX1/2-BUPS1/2	Regulate pollen tube development, Induces pollen tube rupture	[55,56]
CEP3, CEP5	*Arabidopsis*	CEPR1	Regulates plant lateral root growth and the auxin response in lateral root formation	[57,58]
CEP46-D05	Cotton	Unknown	Reduces plant height, fiber length and root length	[59]
IDA/IDL	*Arabidopsis*, *Brassica napus*	HAESA, HSL2	Causes root cap shedding and inflorescence abscission	[60,61]
IDL1	Iitchi	Unknown	Promotes floral organ abscission	[62]

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
