# Peer review of "Small Peptides: Orchestrators of Plant Growth and Developmental Processes"

_ijms, 2024, doi:10.3390/ijms25147627_

Round 1

Reviewer 1 Report

Comments and Suggestions for Authors

The review is a complete and interesting report on a topic that is not widely known. I have enjoyed reading it. I have only two suggestions for increasing the interest of the paper:

a) Figures are very illustrative and nice, but a table sumarizing all the described small peptides, its role and the reference would be very helpful.

 b) Reading the review one has the feeling that most of the research has been performed in Arabidopsis. This casts the doubt or whether most of this findings are general of plants, or particular to plants related to Arabidopsis. Another point, related to this, would be the application in crop plants. Can you include another point and a table summarizing all the spp discovers related to crops and its role/or application. I would also like the auhtors to argue whether they belive this could be biotechnological strategy for crop improvement.

Figure 1 is difficult to read. Please enlarge the lettering.  

Reviewer 2 Report

Comments and Suggestions for Authors

The review of Lu and Xiao entitled “Current Advances in Small Peptides During Plant Growth and Development” summarizes the recent developments in the identification and characterization of small peptides in plants. The review has a lot of information that has to be organized better.

Title

The title has to be revised as it is not written correctly in English. ”Current advances in small peptides …” is just not OK.

Abstract

Line 15

Remove or replace “forward-looking”. It is awkward to say this in English.

Introduction

Lines 49-51

Revise the sentence:

“Using these small peptides as exemplars, the review explores their origins and characteristics, summarizing how they fine-tune plant growth and development by activating various downstream signal transduction pathways.”

Why “examplars”?

“The review explores …… summarizing …” is not OK. Maybe use “… and summarizes…”

Numbering of the sections and subsections is not OK. If introduction was 1,  Classification & Identification of SPs should be 2. Classification Based on Origin should be 2.1 and Based on the N-Terminal Sequence (This should be Classification Based on the N-Terminal Sequence, so, add the word classification) should be 2.2.

Identification Methods of SPs is a completely different topic so it should be 3.

After this a short section should make clear to readers that the next topic that will be discussed is about the most important small peptides. This should be section 4.

Then, Tyrosine-Sulfated Peptides should be 4.1

4.2 PSK Peptides

4.3 PSY Peptides

4.4 CIFs and TWS1

4.5 RGF/GLV/CLEL Peptides

4.6 CLE Peptides

4.7 CLV3

4.8 TDIF

4.9 CLE40

4.10 CLE45

4. 11 Other SPs

4.11.1 EPF/EPFL Peptides

4.11.2 LURE Peptides

4.11.3 RALF Peptides

4.11.4 CEP Peptides

4.11.5 IDA Peptides

5. Conclusions and Perspectives

Other comments.

Are the drawings from figures 1 – 5 original? If yes, the authors could still acknowledge the source of the main information. If not, extensive citations are needed.

Line 63.

The following sentence contains incorrect information: “Most SPs typically have a molecular weight of around 25 amino acids.” Molecular weight is expressed in kilo Daltons (kDa) not in the number of amino acids. This information has to be revised.

Figure 1

Leave a space between proteolytic and cleavage.

Funtional should be corrected to functional.

Line 103

Add “classification” to the title.

Line 115-116

How are the small peptides “traversing the epidermis or xylem”?

Line 125-126

The following sentence needs to be revised:

“Various functional plant SPs have been identified and annotated to regulate multiple 125 aspects of plant growth”.

Why “regulate”?

Figure 2 and lines 161-186

The main paragraphs from lines 161-186 and the legend are not following the order of drawings in figure 2. First it is described what is depicted on the right side of the drawing and then on the left side. This is not OK. The authors should move “NCDS, sORFs, putative SSP-encoding sORFs” to the right of the drawing or change the order on which they present the information in the legend and in the text.

Line 172

Change the word “diminutive” to something more suitable.

Include a short paragraph with a title something like “4. Specific peptides” before Tyrosine-sulfated peptides.

Line 587

Remove “and” before senescence.

Line 589

When indicating “low expression levels” one usually think about mRNA molecules. For small peptides it is better to use “abundance”.

Lines 593-595

In the review the authors described the most recent methods and discussed how complex are the interactions between SPs and various receptors, signaling pathways, functions, and so on.

How is then possible to write something like: “Simplified, accurate, and rapid methods for identifying and verifying their functions are urgently needed.” Is this possible? Can they give us a hint on how these methods should look like or they only wrote something to look  cool? If they cannot than they should remove or rewrite this sentence.

References

The scientific names of plants should be italicized in the reference section.

Comments on the Quality of English Language

 Moderate editing of English language required

Round 2

Reviewer 2 Report

Comments and Suggestions for Authors

The authors addressed my comments.